# The Influence of Adipokines on Radiographic Damage in Inflammatory Rheumatic Diseases

**DOI:** 10.3390/biomedicines11020536

**Published:** 2023-02-13

**Authors:** Eric Toussirot

**Affiliations:** 1INSERM CIC-1431, Centre d’Investigation Clinique Biothérapie, Pôle Recherche, CHU de Besançon, 25000 Besançon, France; etoussirot@chu-besancon.fr; 2INSERM UMR1098 “Relations Hôte Greffon Tumeurs, Ingénierie Cellulaire et Génique”, Université de Franche-Comté, 25000 Besançon, France; 3Rhumatologie, Pôle PACTE (Pathologies Aiguës Chroniques Transplantation Éducation), CHU de Besançon, 25000 Besançon, France; 4Département Universitaire de Thérapeutique, Université de Franche-Comté, 25000 Besançon, France

**Keywords:** adipokines, rheumatoid arthritis, axial spondyloarthritis, bone erosion, syndesmophyte

## Abstract

Inflammatory rheumatic diseases (IRDs) are complex immune-mediated diseases that are characterized by chronic inflammation of the joints. Rheumatoid arthritis (RA) and spondyloarthritis (SpA), including axial SpA (ax SpA) and psoriatic arthritis (PsA), are the most common forms of IRD. Both RA and ax SpA are characterized by a chronic course with progressive structural modifications, namely, cartilage damage and bone erosions in RA and osteoproliferative changes with spinal ossifications in ax SpA. The adipose tissue is involved in the pathophysiology of IRDs via the release of several proteins, namely, adipokines. Several adipokines with pro-inflammatory effects have been identified, such as leptin, adiponectin, visfatin and resistin. In this review, we discuss the role that adipokines may play in the structural modifications of the peripheral joints and/or axial skeleton. In RA, the role of leptin in structural damage remains controversial, while adiponectin and its high-molecular-weight isoform are known to have an influence on the development of bone erosions and radiographic progression. Resistin also appears to be a potent detrimental adipokine for the joints in RA. In ax SpA, visfatin seems to be an attractive candidate for radiographic progression, while leptin and adiponectin have negative effects on radiographic progression.

## 1. Introduction

Inflammatory rheumatic diseases (IRDs) are complex immune-mediated diseases that are characterized by chronic inflammation of the joints and related musculoskeletal anatomical structures. The interaction between genetic and environmental factors is involved in IRDs by initiating and perpetuating complex pathophysiological mechanisms, leading to chronic inflammation with detrimental consequences such as articular damage. The most common IRDs include rheumatoid arthritis (RA) and spondyloarthritis (SpA).

Rheumatoid arthritis (RA) is a systemic immune-mediated disease with typical symmetrical peripheral joint inflammation, joint destruction related to cartilage damage and bone erosion, leading to progressive joint disability and impaired functional capacities. The synovial membrane is the central site of inflammation in RA with a dense infiltrate composed of T and B cells, as well as macrophages and neutrophils. Resident synovial cells, such as macrophage-like and fibroblast-like synoviocytes, are also involved in the local inflammatory response. Multiple pro-inflammatory cytokines, such as IL-1 beta, TNF alpha, IL-6 and IL-17A, and specific chemokines have been identified as key drivers of joint inflammation in RA [1]. Methotrexate is the first line of treatment, but targeted drugs such as biological agents (TNF inhibitors, IL-6 inhibitors, B-cell-depleting agents and costimulatory inhibitors) may dramatically improve the different manifestations of RA. More recently, Janus kinase (JAK) inhibitors have proved to be highly effective in RA [2].

Spondyloarthritis (SpA) refers to a group of inflammatory disorders that mainly involve the axial skeleton, i.e., the spine and sacroiliac joints, but also entheseal structures. Spondyloarthritis may also present with peripheral arthritis, especially of the lower limbs. In this regard, SpA may be classified as axial (ax SpA) or peripheral (pSpA) disease according to the main clinical manifestation. Axial spondyloarthritis is characterized by inflammation in the pelvis and the spine with subsequent and progressive new bone formation. This process may lead to structural modifications of the sacroiliac joints, ultimately leading to sacroiliac joint fusion. New bone formation in the spine corresponds to specific ligamentous ossifications, the development of bony bridges between vertebral bodies and thus progressive ankylosis of the spine with reduced spinal mobility and disability. Radiographic changes in the sacroiliac joints and ligamentous ossifications of the spine are specific hallmarks of ax SpA. TNF alpha and IL-17A have been identified as major players driving inflammation in ax SpA [3]. TNF inhibitors and IL-17 inhibitors have demonstrated their efficacy in ax SpA and are proposed as a second line of therapy after non-steroidal anti-inflammatory drugs (NSAIDs) failure.

Psoriatic arthritis (PsA) is a type of inflammatory arthritis that is typically associated with psoriasis, psoriatic nail disease and/or a family history of psoriasis. Psoriatic arthritis may manifest as peripheral arthritis but also as axial disease and specific musculoskeletal manifestations, such as dactylitis and enthesitis. Psoriatic arthritis belongs to the spectrum of psoriatic diseases but is also included in the group of SpA, with pSpA as its predominant clinical presentation. Both psoriasis and PsA share similar Th1- and Th17-driven inflammation with the increased production of certain inflammatory cytokines, such as TNF alpha, IFN-gamma, IL-6, IL-8, IL-17 and IL-23, in the skin and synovial membrane. Indeed, the IL-23/Th17 pathway has been identified as a critical player in the inflammation of the skin and joints of subjects with psoriasis and PsA. Patients with PsA may experience chronic, progressive disease with both joint erosions and osteoproliferative changes [4]. Methotrexate is usually proposed as the first line of treatment in PsA, and different classes of biological agents are then used in inadequate responders. Biological agents in PsA include TNF inhibitors, IL-17 inhibitors and IL-23 inhibitors, as well as synthetic targeted drugs such as JAK inhibitors.

Adipose tissue is ubiquitously present in the human body and has a wide range of functions. It is a central storage organ with a role as an energy reserve, but it is also involved in metabolic, hormonal, reproductive and immunological functions [5,6]. Adipose tissue, via its dominant cell type, the adipocyte, secretes a wide range of bioactive substances called adipokines. There are numerous lines of evidence showing that, in parallel, adipocytes produce pro-inflammatory factors that are implicated in diverse inflammatory diseases, including IRDs. In this regard, TNF alpha, IL-6 and also growth factors or adhesion molecules may be produced by adipose cells [6]. The most widely described adipokines are leptin, adiponectin, visfatin and resistin. There is now compelling evidence demonstrating the role that adipokines play not only in RA but also in ax SpA or psoriatic disease [7,8,9,10]. Due to the complex interplay that adipokines have not only with immune cells but also with cartilage cells and bone cells, they are also potentially implicated in joint degradation in IRDs [9,10,11,12]. Adipokines such as leptin and adiponectin have been implicated in the pathophysiology of osteoarthritis (OA), a condition with cartilage damage but also new bone formation with the development of osteophytes [10]. Since adipokines are involved in joint/systemic inflammation in IRDs, their effects on joint damage, namely, bone destruction or/and bone formation, are a relevant and important issue [9,11,12,13].

Thus, in this narrative review, we examine the role that adipokines play in structural modifications in IRDs. We focus on the most widely studied molecules in the field, i.e., leptin, adiponectin, visfatin and resistin.

## 2. Methods

In our review, we considered the roles that leptin, adiponectin, visfatin and resistin may have in structural modifications in RA, SpA and PsA. We consulted PubMed to perform this review. The mesh terms used were “adipokines” and “leptin” and “adiponectin” and “visfatin” and “resistin” combined with “bone erosion” and “structural damage” and “osteoproliferation” and “rheumatoid arthritis” and “axial spondyloarthritis” and “psoriatic arthritis”. Only studies published in English were included in this review.

## 3. Adipokines: Main Biological and Immunological Properties

Adipokines are mainly produced by adipocytes but also by other cellular sources, including immune cells, synoviocytes and chondrocytes [5,9,12]. The contribution of adipokines in IRDs has been extensively described, with high levels of adipokines reported in both the blood and the synovial compartment [9,10,11,12,13,14]. The expression of certain adipokines has also been reported in the synovial tissue [10]. The interplay between adipokines and immune cells from both the innate and adaptive systems has been well described, highlighting the role these mediators play in IRDs [9,11,12].

### 3.1. Leptin

Leptin is a 16 kDa hormone mainly produced by adipose tissue. Leptin is primarily involved in appetite control and the energetic balance. However, leptin has a wide range of physiological functions [15]. It strongly correlates with fat mass and body mass index (BMI) and is considered to be a surrogate marker of adiposity. Leptin production is dependent on energetic factors, including insulin and sex hormones, but also pro-inflammatory mediators such as TNF alpha, IL-6 and IL-1 beta, which may also stimulate its release [16,17]. Leptin synthesis is stimulated by ovarian sex hormones and inhibited by testosterone, and thus, circulating leptin levels are more elevated in women than in men, even after adjustment for BMI [15]. The immunomodulatory effects of leptin are well described, and in general, it is considered to be a pro-inflammatory adipokine because of its connection with the innate and adaptive immune systems [16]. Indeed, leptin stimulates the production of pro-inflammatory cytokines such as TNF alpha and IL-6, and alternatively, TNF alpha and IL-1 beta increase the expression of leptin in adipose tissue [17]. Leptin may stimulate monocytes, macrophages, dendritic cells, neutrophils and NK cells [16,17]. Leptin increases the phagocytic activity of monocytes/macrophages, inducing the production of nitric oxide and several cytokines [17]. Leptin is able to induce T-cell activation and differentiation toward a Th1 subset, leading to the increased production of IFN-gamma and IL-2 and the decreased production of IL-4 [16]. Leptin is also associated with Th17 cell proliferation and stimulation [18]. Collectively, leptin is considered to have marked pro-inflammatory properties [13,16].

### 3.2. Adiponectin

Adipose tissue is the main site of adiponectin production, and adiponectin is expressed in different molecular isoforms: globular adiponectin, full-length adiponectin, and low- (LMW), middle- and high-molecular-weight (HMW) adiponectin [9]. Adiponectin has a predominant metabolic function by increasing insulin sensitivity. In fact, patients with insulin resistance and type 2 diabetes (T2D) have low levels of circulating adiponectin. Contrary to leptin, adiponectin levels are decreased in overweight subjects [5]. The influence of adiponectin on the immune system is complex and depends on its different isoforms [8,9,11,12]. Indeed, it is considered that adiponectin has predominant anti-inflammatory effects on atherosclerosis, metabolic syndrome and T2D [19,20]. A favorable effect on cardiovascular (CV) risk has been reported for the specific HMW isoform of adiponectin. In parallel, contradictory effects have been reported in inflammatory disorders such as IRDs [9,11]. Adiponectin may induce the production of anti-inflammatory cytokines such as IL-10 and IL-1 receptor antagonists [21]. TNF alpha and IL-6 may inhibit adiponectin gene expression and protein release. The blockade of the adiponectin receptor AdipoR1 inhibited synovial inflammation in collagen-induced arthritis [22]. In parallel, the different isoforms of adiponectin may exert distinct and sometimes opposing biological functions: LMW adiponectin inhibits LPS-mediated IL-6 release and stimulates IL-10 secretion, while HMW adiponectin induces the secretion of IL-6 by monocytes [23]. In cultured synovial fibroblasts from patients with RA, adiponectin may induce the production of pro-inflammatory mediators [9]. Taken together, it is considered that HMW adiponectin has pro-inflammatory effects on the joint, whereas the LMW isoform is instead associated with anti-inflammatory properties. All in all, adiponectin is considered to have anti-inflammatory effects [19]. However, the opposing effects of adiponectin depend on the relative ratio of its different isoforms, the cytokine environment and the target cell or tissue that is analyzed [11]. In addition, metabolic syndrome (MS) includes different metabolic disorders that increase the risk of diabetes and CV diseases. Visceral fat mass and its derived adipokines may contribute to the development of MS. In this sense, leptin and adiponectin or the leptin/adiponectin ratio may be considered surrogate markers of insulin resistance and MS [24,25].

### 3.3. Resistin

Resistin is a 12.5 kDa cysteine-rich protein that circulates in the blood as a homodimer. It is mainly produced by mononuclear cells and may induce immune cell activation [7,8,9]. Adipose tissue is also a source of resistin production, but to a lesser extent. Resistin is involved in metabolic functions such as insulin resistance [26]. Resistin is found in areas of inflammation and is capable of inducing the production of IL-6, TNF alpha and IL-1 beta by peripheral blood mononuclear cells [27,28].

### 3.4. Visfatin

Visfatin (also known as pre-B-cell colony-enhancing factor or nicotinamide phosphoribosyltransferase) is mainly produced by visceral adipose tissue, a fat tissue localization that strongly correlates with increased CV risk. Visfatin regulates insulin secretion and has insulin-like effects [7,8]. Visfatin has pro-inflammatory effects in various cells via the release of TNF alpha, IL-1 beta, IL-6 and chemokines [29].

## 4. Adipokines in IRD and Their Effects on Joint Components

Adipokines have been extensively studied in patients with RA, but information is less abundant for ax SpA or PsA. The adipokines studied were predominantly leptin and adiponectin and, to a lesser extent, visfatin and resistin [6,7,8,9,10,11,12] (Figure 1).

### 4.1. Rheumatoid Arthritis

Collectively, studies have shown higher circulating levels of leptin, visfatin and resistin in patients with RA and the presence of leptin and other adipokines in the synovial fluid of these patients [30,31,32,33,34,35]. A relationship between disease activity and acute-phase reactants such as CRP and leptin has been described, but with contradictory results [36].

Leptin has pro-inflammatory activity and may potentially influence the process of bone erosion in RA by favoring the release of pro-inflammatory mediators, which in turn may activate resident synovial cells to produce metalloproteinases (MPPs) and receptor antagonist of nuclear factor κB ligand (RANKL) for cartilage and bone degradation, respectively [13]. Indeed, in RA synoviocyte-like fibroblasts, leptin may induce the expression of IL-6 and IL-8 with the involvement of the JAK2/STAT3 pathway [37]. In leptin-deficient mice, the severity of arthritis was reduced, along with TNF alpha and IL-1 beta levels [11]. The relationship between periodontal diseases and RA is well established. In this regard, high levels of circulating leptin were reported in patients with early RA and periodontitis and in those with the presence of *Porphyromonas gingvalis* (*P. gingivalis*) in the oral cavity [38]. In first-degree relatives of patients with RA, serum leptin levels were associated with tender joints, radiographic changes and also the presence of *P. gingivalis* [39]. In addition, leptin may induce increased levels of IL-8 and vascular cell adhesion molecule-1 (VCAM-1) in human chondrocytes [40]. Leptin may induce the gene expression of pro-degradative proteases known as ADAMTSs (ADAMTS4, ADAMTS5, ADAMTS9) in human chondrocytes by activating mitogen-activated protein (MAP) kinases and NFκB [41]. Alternatively, leptin may induce the expression of oncostatin M in osteoblasts from healthy human donors [42], as well as RANKL, bone morphogenic protein (BMP) 4 and alkaline phosphatase in mouse vascular smooth muscle cells, promoting osteoblastic differentiation [43]. Leptin may also reduce the differentiation of bone marrow stem cells into osteoclasts and stimulate the production of osteoprotegerin by mononuclear precursors, favoring an osteoblast lineage and thus osteogenesis [44].

A number of studies have reported that adiponectin levels are increased in the serum and in the synovial fluid of patients with RA compared to healthy controls or patients with OA [31,45,46,47,48,49,50,51,52]. Adiponectin levels were found to correlate with the disease activity of RA in certain studies. The synovial tissues of patients with RA, especially fibroblast-like synoviocytes, expressed adiponectin and its receptors, AdipoR1 and AdipoR2 [53]. In addition, in an experimental model, adiponectin alone or in combination with IL-1 beta may induce the release of IL-6, IL-8 and PGE2 by fibroblast synoviocytes [54]. The metalloproteinases MMP-1 and MMP-3 are also produced by fibroblast-like synoviocytes when stimulated by adiponectin [21]. Adiponectin may activate osteoclastic activation through the stimulation of RANKL and the inhibition of osteoprotegerin production by osteoblasts, thus favoring joint bone erosion [8,13]. In human chondrocytes, adiponectin stimulated the production of IL-6, IL-8, MMP-3, MMP-9 and monocyte chemoattractant protein-1 (MCP-1) [55]. It has been reported that HMW adiponectin is likely to induce a stronger pro-inflammatory reaction in RA synovial fibroblasts compared to the other isoforms [11]. Indeed, HMW adiponectin induced the secretion of IL-6 in human monocytes, while LMW adiponectin reduced IL-6 release and stimulated IL-10 secretion by LPS-activated monocytes [56]. Adiponectin is able to promote the differentiation of naïve CD4+ T lymphocytes toward a Th17 phenotype, thus contributing to joint inflammation and bone erosion. This differentiation was dependent on the AdipoR1 receptor, as demonstrated by an AdipoR1 knockout model [57]. The intra-articular injection of adiponectin in the joints of mice with collagen-induced arthritis resulted in severe disease with marked inflammation and synovial hyperplasia, RANKL expression and bone erosions [58]. Finally, it was demonstrated that the stimulation of RA synovial fibroblasts by adiponectin induced the production of osteopontin, which in turn recruited osteoclasts to the bone surface to initiate bone erosions [59]. Collectively, these data effectively demonstrate the degradative properties of adiponectin on the joint components.

Serum concentrations of visfatin were found to be higher in patients with RA, compared to healthy controls or patients with OA [31,60]. Visfatin is positively correlated with measurements of disease activity of RA [31]. Visfatin is able to induce a pro-inflammatory response in various cells, including RA fibroblast-like synoviocytes, monocytes, chondrocytes and bone cells [11]. For instance, visfatin activated human chondrocytes to produce pro-inflammatory cytokines, MMP-3 and PGE2 [12,13]. Osteoblasts may respond to visfatin by secreting IL-6 and chemokines, while conflicting results have been observed for osteoclasts, with both inhibition and activation of their differentiation [61,62]. Visfatin may potentially promote angiogenesis by increasing RA synovial fibroblast adhesion to endothelial cells under static and flow conditions [63]. In an animal model of arthritis, osteoclastogenesis required the presence of visfatin, suggesting a direct role in joint damage [64].

Resistin has been found to be moderately elevated in the sera of patients with RA, while its concentration is markedly increased in the synovial fluid compared to results obtained from subjects with OA [28,65]. A meta-analysis of eight studies of RA reported that serum resistin was higher in patients with RA compared to normal controls [66]. Resistin is expressed in the synovial fluid and the synovial membrane of patients with RA. Local cells in the joint cavity, including RA synovial fibroblasts, macrophages, B cells and osteoblasts, expressed resistin [67]. Serum resistin levels correlated with systemic biomarkers of inflammation, such as CRP or the erythrocyte sedimentation rate [68]. When stimulated by resistin, RA synovial fibroblasts produced pro-inflammatory factors and chemokines [69]. Resistin may also promote angiogenesis among endothelial progenitor cells by increasing vascular endothelial growth factor (VEGF) production [11].

### 4.2. Axial Spondyloarthritis

More limited information is available on adipokines in ax SpA. Conflicting results have been reported for circulating leptin levels in ankylosing spondylitis (AS) (also called radiographic ax SpA), with decreased levels in some studies [70,71] and increased levels in another [72]. Serum leptin correlated with the Bath Ankylosing Spondylitis Disease Activity Index (BASDAI), a validated patient-reported index of disease activity, and with acute-phase reactants (CRP, IL-6) [73]. The relationships between adipokines and the ankylosing spondylitis disease activity score (ASDAS), a more recent index of disease activity including acute-phase reactants, have never been examined. In a recent meta-analysis, it was reported that there was no difference in serum leptin or serum adiponectin between AS and controls, while patients with AS had higher serum resistin levels [74]. These results for serum leptin were confirmed by a second meta-analysis, indicating that there is no significant difference in plasma/serum leptin between patients and controls [75]. Certain adipokines have the potential to stimulate bone cells toward an osteogenic profile and/or to activate osteoblasts to induce bone formation, a relevant question for ax SpA. In this sense, visfatin is able to bind to osteoblasts, stimulate their proliferation and induce chemokine production and type X collagen expression [76,77]. Leptin has previously been linked to bone remodeling and thus has been implicated in the pathophysiology of osteoporosis [78]. However, the effects of leptin on bone formation remain debated, with both positive and negative consequences [79]. Indeed, leptin has effects on the central nervous system and the peripheral regulation of bone cells. On the one hand, at the central level, leptin has complex hormonal, autonomic nervous system and hypothalamic interactions, resulting in bone loss [80]. On the other hand, the peripheral effects of leptin result from direct interactions with bone cells: leptin may activate osteoblasts and inhibit osteoclasts [81,82]. Adiponectin and its receptors are expressed in osteoblasts and can stimulate their activation and differentiation. Thus, adiponectin may promote osteoblastogenesis and suppress osteoclastogenesis [83]. Conversely, adiponectin may activate osteoclasts by producing RANKL and inhibiting osteoprotegerin in osteoblasts [84]. The influence of resistin on bone remodeling is less well established. Indeed, resistin is expressed in human osteoblasts and can increase the osteoclast number and activity, but it also has a weak effect on osteoblasts by enhancing their proliferation. These data indicate a global impact of resistin on both osteoclastogenesis and osteoblastogenesis [85].

### 4.3. Psoriatic Arthritis

There are limited data on adipokines in patients with PsA [86,87]. High serum levels of leptin and low levels of adiponectin in patients with PsA compared to healthy controls have been reported [88]. In a large Canadian series, adipokines were compared between patients with psoriasis alone and patients with PsA. The results showed higher adiponectin levels in patients with PsA and higher levels of leptin in PsA, but only among women [89]. Finally, circulating leptin, adiponectin and resistin levels have been found to be higher in PsA than in healthy subjects [90]. The relationship between these adipokines and the levels of disease activity of PsA was not clear, either with laboratory parameters of inflammation or with clinical measurements of disease activity [88].

## 5. Adipokines and Structural Modifications in IRDs

Inflammatory rheumatic diseases are characterized by the development of structural joint damage, with cartilage degradation and bone erosions in RA and new bone formation and ossifications in ax SpA. Psoriatic arthritis may combine both erosions and osteoformation at the level of the joints.

### 5.1. Rheumatoid Arthritis

Leptin was the first adipokine to be evaluated for its influence on joint damage [91,92]. The serum leptin concentration was found to be higher in patients with erosive disease compared to those with non-erosive RA [93]. In a study evaluating leptin in paired synovial and blood samples, leptin was higher in the plasma than in the synovial fluid, with a greater difference between levels of leptin in the plasma and the synovial fluid in patients with non-erosive RA than in patients with erosive RA [91]. These results suggested that the local penetration of leptin into the joint cavity may have a protective effect against the development of erosion. Alternatively, leptin may have a role in the erosive process. Different studies have examined the question of the relationship between serum leptin and radiographic joint damage in RA, giving contradictory results (Table 1) [46,47,48,49,94,95]. In four studies, serum leptin did not correlate with joint damage [47,48,49,91], while two other studies found that serum leptin levels correlated with the radiographic joint score: there was an inverse relationship between leptin levels and the Larsen score (Odds Ratio (OR): 0.32 [95% CI: 0.17–0.62]) in one study [46], while the progression of the radiographic joint score was independently associated with serum leptin levels in a second study (OR: 1.59, 95% CI: 1.05–2.42) [94]. Collectively, the potential role of leptin on joint structural damage in RA is thus not currently elucidated.

Different studies have analyzed the potential role of adiponectin in structural damage in RA. Concordantly, these studies showed that adiponectin was associated with the joint erosive process and the progression of the disease (Table 1) [47,48,49,94].

In one study, visfatin concentrations were correlated with the radiographic Larsen score, independently of age, sex, disease duration, BMI and inflammation (OR: 2.38, 95% CI: 1.32–4.29) [46]. This was confirmed by a second study, which reported higher visfatin levels in patients with erosive disease compared to patients without radiographic erosions, but there was no correlation between the Larsen score and visfatin serum concentrations [95].

Finally, resistin was identified as a predictor of radiographic progression at 5 years in a Finnish cohort of 99 RA patients participating in the NEO-RACo trial [96].

### 5.2. Axial Spondyloarthritis

The role of adipokines on the radiographic structural progression of the spine has been examined in different studies. Certain studies reported that serum leptin was higher in patients with ligamentous ossifications compared to patients without [97,98], while another study reported a relationship between changes in serum leptin at 2 years and radiographic progression, as evaluated by the modified stoke ankylosing spondylitis spinal score (mSASSS) [99] (Table 2). Conversely, two studies reported an inverse relationship between serum leptin and radiographic progression at 2 years in the first study (OR for the outcome “no mSASSS progression”: 1.16 [1.03–1.29]) [100] and at 4 years for the second (OR for the outcome “mSASSS progression”: 0.614 [0.453–0.832]) [101]. In parallel, visfatin (but not resistin or adiponectin) has been linked to the worsening of the radiographic mSASSS score at 2 years [102]. These results were confirmed when analyzing another cohort of AS patients: the change in visfatin levels was associated with mSASSS progression at 4 years (OR: 2.255 [1.108–4.581]) [101]. Finally, HMW adiponectin was inversely correlated with spinal radiographic progression [100].

### 5.3. Psoriatic Arthritis

There is currently a limited body of data regarding the role of leptin, adiponectin or other adipokines in structural joint damage in patients with PsA. In one study involving 41 patients with PsA, high serum leptin and low serum adiponectin levels were found in the PsA group, but without any correlations with the radiographic damage score (Table 3). On the contrary, leptin positively correlated with the number of osteoclast precursors [88].

## 6. Conclusions

Adipokines are a complex group of molecules with pleiotropic functions, interacting with immune and bone cells. They are effective contributors to the inflammatory reaction in the joint cavity and, therefore, have the potential to participate in the process of joint damage in IRDs.

In RA, they can promote cartilage damage and bone erosions, but their exact contributing role is probably less evident regarding the marked effects of pro-inflammatory cytokines such as IL-1 beta, TNF alpha, IL-6 or IL-17A. Due to their pro-inflammatory activities, adipokines can stimulate various cells in the joints, including macrophage and fibroblast-like synoviocytes. They can thus be associated with the production of deleterious mediators such as MMP or RANKL, thus promoting joint component degradation. The role of leptin in RA structural damage seems controversial, while the influence of adiponectin and its HMW isoforms is concordant, promoting bone erosion and radiographic progression. Resistin also appears to be a potent detrimental adipokine for the joints.

In ax SpA, the opposite context is described, with a bone formation mechanism and the development of osteoproliferative changes in the spine. In this regard, visfatin seems to be an attractive candidate, but its specific role in promoting syndesmophyte formation with regard to the well-identified predisposing factors for radiographic progression (such as male sex, CRP, smoking, pre-existing syndesmophytes) remains to be clarified. On the contrary, leptin and adiponectin seem to be negative factors for radiographic progression.

In PsA, data on the role of adipokines in bone structural damage are very preliminary and require future studies.

Some limitations of these results must be discussed, such as the influence of BMI (or correction of leptin for fat mass), the socio-demographics of the included patients, but also the method chosen to evaluate the radiographic damage (radiographic score). In addition, adipokines are biomarkers implicated in both inflammation and structural changes in IRD. As has been demonstrated in patients with ax SpA [103], they must be considered as additional factors on top of other biological parameters, such as acute-phase reactants, cytokines, chemokines or MMPs, as well as clinical parameters, in the understanding of structural damage/radiographic progression in IRDs.

## Figures and Tables

**Figure 1 biomedicines-11-00536-f001:**
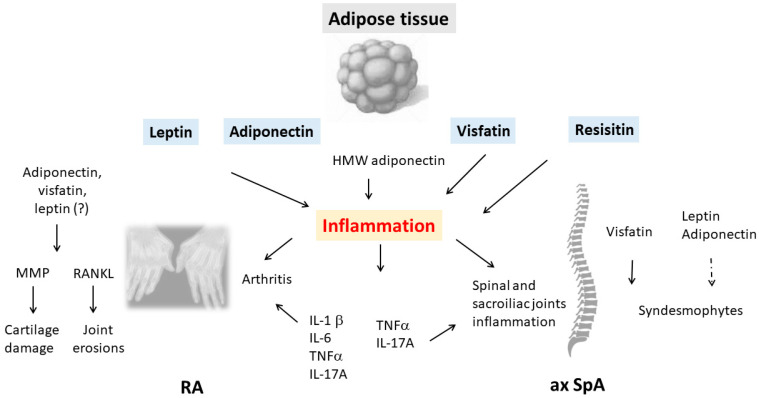
The contribution of adipokines to joint modification in inflammatory rheumatic diseases (IRDs). Several factors are produced by the adipose tissue, collectively called adipokines. Leptin, adiponectin, visfatin and resistin are all involved in inflammation of the joints/axial skeleton in rheumatoid arthritis and axial spondyloarthritis. By stimulating pro-inflammatory cytokine release from innate and adaptive immune cells, adipokines induce an inflammatory milieu. In turn, pro-inflammatory cytokines may promote the release of adipokines. Leptin, visfatin and resistin all have pro-inflammatory effects on the joints and/or axial skeleton, while the effects of adiponectin on the inflammatory process are variable and depend on its isoform, with the high-molecular-weight isoform driving a pro-inflammatory reaction. Due to the interplay that adipokines have with specific factors involved in cartilage and bone metabolism, they may contribute to cartilage degradation and bone erosions in RA or spinal ligamentous ossifications in ax SpA (RA: rheumatoid arthritis; ax SpA: axial spondyloarthritis; HMW: high molecular weight; MMP: metalloproteinases; RANKL: receptor activator of nuclear factor κB ligand; dashed arrow: inhibitory effect; lined arrow: stimulating effect).

**Table 1 biomedicines-11-00536-t001:** Relationships between adipokines and radiographic damage in rheumatoid arthritis.

Author, Ref.	No. of Pts	Sex Ratio (M/F)	Disease Duration (Years)	Treatment	Evaluated Adipokines	Structural Damage Assessment	Main Results
Bokarewa [91]	76	28/48	11	Predominance of csDMARDs Only 5 patients under TNFi	Leptin in the plasma and synovial fluid	Presence/absence of erosion	↑ Leptin plasma; ↓↓ Synovial; fluid/controls; Leptin synovial fluid with erosive disease > non-erosive disease
Giles [47]	197	79/118	9	csDMARD: 84.2% bDMARD: 45.4%	Total adiponectin Leptin Resistin	SHS	Association between serum adiponectin and joint damage; Leptin and resistin were not related to joint damage
Rho [46]	167	52/115	3	csDMARDs: 90% bDMARDs: 25%	Leptin Total adiponectin, Resistin, Visfatin	93 patients evaluated by Larsen score	Visfatin associated with Larsen score; Elevated leptin levels associated with reduced joint damage
Giles [48]	Cohort of 152 patients followed for 40 months	57/95	9	csDMARDs: 86% bDMARDs: 44%	Total adiponectin Leptin Resistin	SHS	Association between higher adiponectin levels and progression of structural damage
Klein-Wieringa [49]	253 patients followed for 4 years	79/174	Early RA (<1 year)	ND	Leptin Total adiponectin Resistin Visfatin	SHS	Total adiponectin associated with radiographic progression over 4 years independently of anti-CCP antibodies
Meyer [94]	632	140/492	Early RA (0.7 year)	No cs/bDMARDs	Leptin Total adiponectin Visfatin	Modified Van der Heijde Sharp score	Total adiponectin associated with SHS score; Adiponectin associated with radiographic progression at 1 year; Leptin associated with radiographic progression
Mirfeizi [95]	54	11/43	3.5	ND	Leptin Visfatin	Larsen score	Visfatin higher in erosive vs. non-erosive disease; No correlation between visfatin and Larsen score
Vuolteenaho [96]	90	29/61	Early RA (0.4 year)	Combination of csDMARDs or csDMARD + infliximab	Resistin	SHS score	High resistin predicted radiographic progression at 2 and 5 years in patients under csDMARDs

RA: rheumatoid arthritis; M: male; F: female; csDMARDs: conventional synthetic disease-modifying anti-rheumatic drugs; bDMARDs: biological disease-modifying anti-rheumatic drugs; TNFi: TNF inhibitor; SHS: Modified Van der Heijde Sharp score.

**Table 2 biomedicines-11-00536-t002:** Relationships between adipokines and radiographic progression in axial spondyloarthritis.

Author, Ref.	No. of Pts	Sex Ratio (M/F)	Disease Duration (Years)	Treatment	Evaluated Adipokines	Structural Damage Assessment	Main Results
Syrbe [102]	86 AS patients	56/30	4.6	NSAIDs: 55% csDMARDs: 29% TNFi: 2.4%	Adiponectin Resistin Visfatin	mSASSS	No association between baseline serum adipokines and mSASSS; Visfatin elevated in patients with radiographic progression at 2 years
Gonzalez-Lopez [98]	48 AS patients	30/18	9.4	csDMARDS: 94% TNFi: 29% Corticosteroids: 15%	Leptin Total adiponectin	Presence or absence of syndesmophytes without scoring	Increased leptin levels in patients with syndesmophytes
Hartl [100]	120 AS patients	82/38	14.8	NSAIDS (continuous or on demand)	Total and HMW adiponectin Leptin Resistin Visfatin	mSASSS	Decreased HMW adiponectin and leptin in patients with radiographic progression; Leptin and leptin/BMI ratio inversely associated with radiographic progression; HMW adiponectin/total adiponectin inversely associated with radiographic progression
Park [99]	20 AS patients	20	4.3	NSAIDs: 100% csDMARDs: 95% Corticosteroids: 5%	Leptin Total adiponectin Resistin	mSASSS	Baseline resistin positively correlated with changes in mSASSS at 2 years; Changes in leptin/BMI ratio associated with radiographic progression
Rademacher [101]	137 AS patients	98/39	15	NSAIDs	Leptin HMW adiponectin Visfatin	mSASSS	No significant differences in baseline adipokine levels between patients with and without radiographic progression Visfatin associated with mSASSS progression at 2 years; Changes in visfatin associated with radiographic progression Leptin inversely associated with radiographic progression at 2 years

AS: ankylosing spondylitis; NSAIDs: non-steroidal anti-inflammatory drugs; HMW: high molecular weight; mSASSS: modified stoke ankylosing spondylitis spinal score; BMI: body mass index.

**Table 3 biomedicines-11-00536-t003:** Relationships between adipokines and radiographic damage in psoriatic arthritis.

Author, Ref.	No. of Pts	Sex Ratio (M/F)	Disease Duration (Years)	Treatment	Evaluated Adipokines	Structural Damage Assessment	Main Results
Xue [88]	41	26/15	3.5	NSAIDs: 80% csDMARDs: 54%	Leptin Resistin Adiponectin	SHS modified for PsA	No differences in serum leptin, resistin or adiponectin between patients with and without erosions Leptin not correlated with radiographic score Leptin correlated with osteoclast precursor number

PsA, psoriatic arthritis.

## Data Availability

No new data were created or analyzed in this study. Data sharing is not applicable to this article.

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
