# Peer review of "The Influence of Adipokines on Radiographic Damage in Inflammatory Rheumatic Diseases"

_biomedicines, 2023, doi:10.3390/biomedicines11020536_

Round 1

Reviewer 1 Report

The present article reviews the correlation between adipocytokines and structural damage in IRDs.  The topic can be considered interesting, but it must be developed more;  few improvements are also needed, which are listed below:

1.   The presentation of information is much too brief for/considering the generosity of the topic.Many aspects should be more developed.

2. The first paragraph of the introduction lacks bibliographical references. If the first two paragraphs belong to [1], then this is a lot of information provided on the basis of a single bibliographic source. Please revise the whole manuscript ([2],[3]-long paragraphs, a single bibliographic source etc.) from this point of view.

3.   RA, SpA, PsA represent complex pathologies with immunological mechanisms, so some aspects related to their pathophysiology/risk factors should be detailed, but also briefly presented the current therapeutic options for an overview of the work and to identify the aspects influenced by adipokines, especially as correlations between adipokines and radiographic damage in different groups of patients under different conventional treatments have been presented in tables. I suggest checking and referring to: PMID: 36058148; PMID: 34831081.

4.   The aim of the study should be better detailed and improved for a better understanding of the motivation of the choice of the study topic, and especially the presentation of the contributions made to the scientific literature.

5.   The information is presented in a relevant and logical manner in terms of the flow and order of the information described. It would therefore be advisable to present the methodology for selecting bibliographic resources (databases used, types of documents, filtering results, etc.).

6.   It is recommended that the figure be mentioned/annotated in the text before being inserted for better clarity of information.

7.   It has been noted that leptin correlates strongly with fat mass and BMI and is considered a surrogate marker of adiposity, and leptin production depends on metabolic and energetic factors. Therefore, it would be advisable to present the correlations between adipocyte biomarkers and metabolic syndrome/insulin resistance based on studies. I suggest checking and referring to: PMID: 33123227; PMID: 32509004.

Author Response

Thank you for giving us an opportunity to revise our manuscript. We would like to express our sincere appreciation for the reviewers' constructive and positive comments and detailed suggestions. We have studied the comments carefully and revised our manuscript accordingly. 
Following are the point-to-point Responses to the reviewers' comments

Sincerely 

  1. The presentation of information is much too brief for/considering the generosity of the topic.Many aspects should be more developed.
  2. The first paragraph of the introduction lacks bibliographical references. If the first two paragraphs belong to [1], then this is a lot of information provided on the basis of a single bibliographic source. Please revise the whole manuscript ([2],[3]-long paragraphs, a single bibliographic source etc.) from this point of view Answer: see below
  3. RA, SpA, PsA represent complex pathologies with immunological mechanisms, so some aspects related to their pathophysiology/risk factors should be detailed, but also briefly presented the current therapeutic options for an overview of the work and to identify the aspects influenced by adipokines, especially as correlations between adipokines and radiographic damage in different groups of patients under different conventional treatments have been presented in tables. I suggest checking and referring to: PMID: 36058148; PMID: 34831081. Answer: thanks for the reviewer comment. For each IRD, we add a short section on the treatment given (methotrexate, biological agents, synthetic targeted agents). Ref   PMID: 34831081 was added but not PMID: 36058148 since it is out of the topic of the paper.
  4. The aim of the study should be better detailed and improved for a better understanding of the motivation of the choice of the study topic, and especially the presentation of the contributions made to the scientific literature.Answer: yes we agree and modify the last section of the introduction
  5. The information is presented in a relevant and logical manner in terms of the flow and order of the information described. It would therefore be advisable to present the methodology for selecting bibliographic resources (databases used, types of documents, filtering results, etc.).Answer: thanks to the reviewer for his/her pertinent remark. We add a section "methods"
  6. It is recommended that the figure be mentioned/annotated in the text before being inserted for better clarity of information.Answer: yes the figure is cited in the paper line 212
  7. It has been noted that leptin correlates strongly with fat mass and BMI and is considered a surrogate marker of adiposity, and leptin production depends on metabolic and energetic factors. Therefore, it would be advisable to present the correlations between adipocyte biomarkers and metabolic syndrome/insulin resistance based on studies. I suggest checking and referring to: PMID: 33123227; PMID: 32509004. Answer: thanks to there reviewer. We add a short section on metabolic syndrome and its interrelation with adipokines. Suggested references were added. 

Reviewer 2 Report

Dear author,

This is an excellent overview of adipokines in general, in RA/SPA/PSA, and radiographic damage in particular. Overall, I do not have any large concerns content-wise.

It would be a nice addition to add some limitations/confounding factors when researching adipokines in IRD.

For example, when doing clinical research what factors can influence results? Eg. BMI, Rx damage score, ... Moreover, how does the heterogeneity of diseases such as PsA and Spa influence this story? Are there other signals in related immune diseases such atopic dermatitis or IBD?

I believe with some extra insights in these matters, this is a good review.

Author Response

Thank you for giving us an opportunity to revise our manuscript. We would like to express our sincere appreciation for the reviewers' constructive and positive comments and detailed suggestions. We have studied the comments carefully and revised our manuscript accordingly. Following are the point-to-point Responses to the reviewers' comments

Sincerely

This is an excellent overview of adipokines in general, in RA/SPA/PSA, and radiographic damage in particular. Overall, I do not have any large concerns content-wise. It would be a nice addition to add some limitations/confounding factors when researching adipokines in IRD. For example, when doing clinical research what factors can influence results? Eg. BMI, Rx damage score, ... Moreover, how does the heterogeneity of diseases such as PsA and Spa influence this story? Are there other signals in related immune diseases such atopic dermatitis or IBD? I believe with some extra insights in these matters, this is a good review.

Answer: Yes and thanks to the reviewer. We add a comment in the conclusions section for limitations/confounding factors

Round 2

Reviewer 1 Report

The authors responded to my requests.

Author Response

-